# High Gain Slot Array Antenna at 110 GHz Based on Computer Numerical Control

**DOI:** 10.3390/mi14101947

**Published:** 2023-10-19

**Authors:** Zhen Tan, Yun Zhao, Jiangqiao Ding

**Affiliations:** School of Electronic & Information Engineering, Nanjing University of Information Science and Technology, Nanjing 210044, China; 202083270489@nuist.edu.cn (Z.T.); jqding@nuist.edu.cn (J.D.)

**Keywords:** high gain antenna, slot array antenna, millimeter wave

## Abstract

This paper presents a waveguide–slot antenna to generate a radiation beam with high gain fed by a low-loss feeding network at 110 GHz. The proposed antenna consists of a compact eight-way power divider and a waveguide–slot array. The eight-way power divider provides equal-amplitude and alternative-phase excitation for the slot array, and each of them supports two waveguides. The integral structure is implemented by two layers with a channeled substratum and a slotted superstratum. To verify the proposed slot array, the designed array is fabricated with computer numerical control (CNC) milling and measured. The measured peak gain of the designed antenna is 32 dBi at 110 GHz. The proposed antenna with a simple structure provides a promising solution to develop high gain antenna in upper millimeter-wave and sub-terahertz (THz) applications.

## 1. Introduction

Millimeter-wave and sub-terahertz (THz) technologies have recently drawn greater attention due to their feasible applications in the areas of high-resolution imaging, radio astronomy, communication systems, etc. [1,2,3]. For most high-frequency applications, high-gain antennas are required to compensate for the electromagnetic wave propagation loss. Therefore, high-gain antennas have been strongly developed in response to the demand for the applications in high-frequency band.

Various studies of high-frequency antenna designs, mostly in the millimeter band, have been conducted for the characteristics of having a high gain, low cost, and strong integrability and being easy to reproduce. Microstrip antennas are typically advantageous in terms of the cost of production and their lightweight features. However, transmission loss increases especially in high-frequency applications [4]. The traditional method for obtaining high-gain antenna is introducing lens. The on-chip antenna with a lens integrated and metallic reflective cavity loaded at 140 GHz was proposed in [5]. However, the lenses are usually large and bulky and suffer from inevitable dielectric losses. A W-band coplanar waveguide fed tapered slot antenna based on rectangular-grooved silicon substrate was proposed in [6]. Although the bandwidth of the antenna was over the whole W band, the gain of the antenna was below 10 dBi because of the obvious electromagnetic wave loss. In order to restrain the electromagnetic wave losses of microstrip antennas, substrate integrated waveguide–slot array antennas were suggested. A SIW slot array antenna was discussed in [7]. It was found that the SIW structure did not have a significant effect on the gain enhancement. The low-temperature cofired ceramic (LTCC) technology is introduced to construct low-loss structures [8,9]. An 8 × 8 antenna with a LTCC-based gap waveguide feeding network was designed in [8]. On the basis of multilayered LTCC technology, the construction of the antenna was complicated, and the gain was ultimately 23.8 dBi due to dielectric loss and energy leakage in the feeding network.

Simple and typical hollow waveguide–slot array antenna structures are significantly more efficient with low transmitting loss and high gain than other types of antennas, even if there is a narrower bandwidth for impedance [10,11,12]. TE_220_ mode in substrate-integrated cavity was excited by the microstrip-based ridge gap waveguide to reduce the transmission loss in [10]. Nevertheless, the gap waveguide feeding network requires numerous routes and patches for a large-scale array, which increases the complexities of antenna design and simulation, while the antenna gain is relatively low. A 32 × 32-element slot array antenna consisting of three layers utilizing chemical-plated polyetherimide in the E-band was reported in [11]. The operating bandwidth covered the 71–86 GHz frequency band by adding the metal brick in the backed cavity, and the gain was 37 dBi. Micromachine technology is developed in recent years and more suitable for higher-frequency applications but with a high cost and rather low gain in [13].

The slotted waveguide array is attractive due to its compact size, planar form and ease to be fabricated based on the computer numerical control (CNC) milling technology [14]. Due to the low transmission loss feature, the traditional metallic waveguide–slot antenna arrays are applied in microwave-millimeter wireless communication systems. This paper presents a design of a waveguide–slot array antenna producing a radiation pattern with narrow bandwidth and high gain which fulfills the requirement of the systems used in human body imaging security detectors at 110 GHz. This antenna is composed of a low-loss power divider and waveguide–slot array. The high-gain slot array antenna is fabricated with copper and tested. Measured results show good agreement with simulated ones. Due to its extremely easy construction and high gain beyond 30 dBi at 110 GHz, the antenna can be potentially very useful in systems at higher frequencies.

## 2. Description of the Slot Array Antenna

Figure 1 illustrates the designed slot array antenna with a symmetric structure along the *y*-axis, which is composed of two layers. The substratum layer is a 16-element waveguide fed by a power divider as shown in Figure 1b, and the superstratum layer is a plate with a slot array as shown in Figure 1c. The length of the slot is *l*_s_, the width *w*_s_, the offset with respect to the central line is *g*_3_, and the slot spacing is *d*_s_. In order to reduce the loss of the electromagnetic propagation in high frequency and easily integrate the whole structure, the feeding network and the slotted waveguide are designed in a plane. Computer numerical control (CNC) milling technology is employed to process the designed antenna metal mechanical processing, which also lowers the final cost.

## 3. Design of the High-Gain Slot Array Antenna

### 3.1. Feeding Network

The high-gain slot array antenna is fed by an eight-way power divider and every two waveguides are connected to one port to obtain a compact construction, as shown in Figure 1. The width of the waveguide is *w*_1_, the gap between adjacent waveguides is *g*_2_, and the width and depth of the rectangular grooves are *g*_1_ and *l*_1_, respectively. The electromagnetic energy of each port is evenly divided into two ways supplied to individual waveguides. According to Figure 2, S_11_ varies below minus 10 dB in the range from 104 GHz to 117 GHz. It can also be seen that the power is evenly distributed to each waveguide in the same frequency band. The feeding network combines a third-order power divider and a compact power divider because of the tightly arranged waveguide structure. Actually, the power divider is fourth-order, so the insertion loss is around 12 dB.

### 3.2. Slot Array Antenna

In this design, a waveguide–slot antenna is chosen as the array element. The rectangular waveguide cavity is enclosed by two layers, one layer with 16 grooves working as a waveguide cavity and the other one etched with 16 groups of slots, which plays the role of the lid of the cavity. This configuration ensures the waveguide is split along its E-plane and is less prone to excessive losses at the split planes.

Linear short-circuited waveguide–slot antenna array with longitudinal slot is a segment with the slots etched in the broad wall of the waveguide (Figure 1). The antenna is designed to work at 110 GHz with strong directivity. Based on the principle of a waveguide–slot antenna, the initial dimensions of the slot are calculated. Slots are spaced apart at a distance of half waveguide wavelength *λg*. Slot length is approximately half free space wavelength *λ*. Distance from the short-circuited end to the last slot is three quarters of the waveguide wavelength. The slot offset *g*_3_ relative to the center line of the waveguide can be varied to obtain the desired characteristics of the antenna array. To simplify the design procedure, the offset of the slots is with a consistent distance of 0.2 mm. The radiation portion is connected to the power divider to constitute the whole high-gain waveguide–slot array antenna. The thickness of the side of the excitation port of the antenna is thicker to load the ring flange and then transmitted to the thickness of the waveguide–slot antenna gradually, as shown in Figure 1a. It is noticed that the thickness of the other three borders of the designed antenna is increased to keep the slotted plane from deforming and accommodate the set screws.

## 4. Measurement and Discussion of the High-Gain Slot Array Antenna

The proposed antenna is fabricated with CNC processing, and the photo of the machined antenna is shown in Figure 3. The overall size of the designed antenna is 45 mm × 61 mm. The completed structure is constituted by two parts with a slotted superstratum and grooved substratum splitting along the median plan of the cavity. The elements of the waveguide–slot array antenna are allocated with equal power using a power divider. The entire structure is dissected along the center line of the narrow side of the excitation port, as shown in Figure 3. Around the border, dowel pins are distributed to increase tooling accuracy.

The fabricated antenna is tested in the environment of a compact range, as shown in Figure 4. Figure 5 shows the simulated and measured S-parameter of the proposed slot array antenna. From the illustration, it can be seen that S_11_ of the simulation result is lower than −10 dB in the frequency range from 108.7 GHz to 111.2 GHz. The measured S_11_ shows good agreement with the simulated results, ignoring the slight deterioration compared with the simulated one due to the fabrication process and test environment.

The simulated and measured radiation patterns of the proposed antenna are shown in Figure 6. It can be seen from the results that the radiation beams in the E-plane are narrower than that in the H-plane because of the broader radiate aperture of the E-plane. The simulated gain over the frequency range is from 32 to 34 dBi, and the measured one is from 31 to 32 dBi. The slight difference between them can be attributed to the mounting error and fabrication tolerance. Although the slots are distributed uniformly, avoiding a complicated calculation of offset based on Chebyshev synthesis or Taylor synthesis, the side lobe level is beyond 20 dBi.

Table 1 provides a fractional gain comparison between our designed antenna and the available various antennas working at millimeter wave reported so far. Generally, all reported antennas require complicated constructions. It is seen that although the antenna in [10] can maintain a slightly higher gain than our proposed antenna, the volume is significantly larger, and the structure is more complicated with a much higher finished cost. The antennas proposed in [7,8] have a much lower gain although with a complicated process. The designs in [6,9,11,14] had a smaller structure but with a lower gain. Micromachine technology is mainly suitable for higher-frequency applications but with a high cost. It is important to point out that, under the requirement of a narrow bandwidth antenna, our proposed antenna has significant advantages, taking all the properties into consideration, including the radiation properties and fabrication process. The manufacturing technology applied in this work has a much lower cost for massive production in industry.

## 5. Conclusions

In this paper, a high-gain slot array antenna has been presented using numerical control machining avoiding a complex and expensive manufacturing process. The proposed antenna consists of a power divider and a waveguide–slot array. The antenna is designed working at a frequency of 110 GHz with a high gain of 32 dBi. A prototype of the proposed waveguide–slot array antenna has been fabricated and tested. The measured results show good agreement with the simulated ones. The proposed antenna shows advantages of a low profile, high gain, and simple construction for easy fabrication, exhibiting powerful potency even in high-frequency applications.

## Figures and Tables

**Figure 1 micromachines-14-01947-f001:**
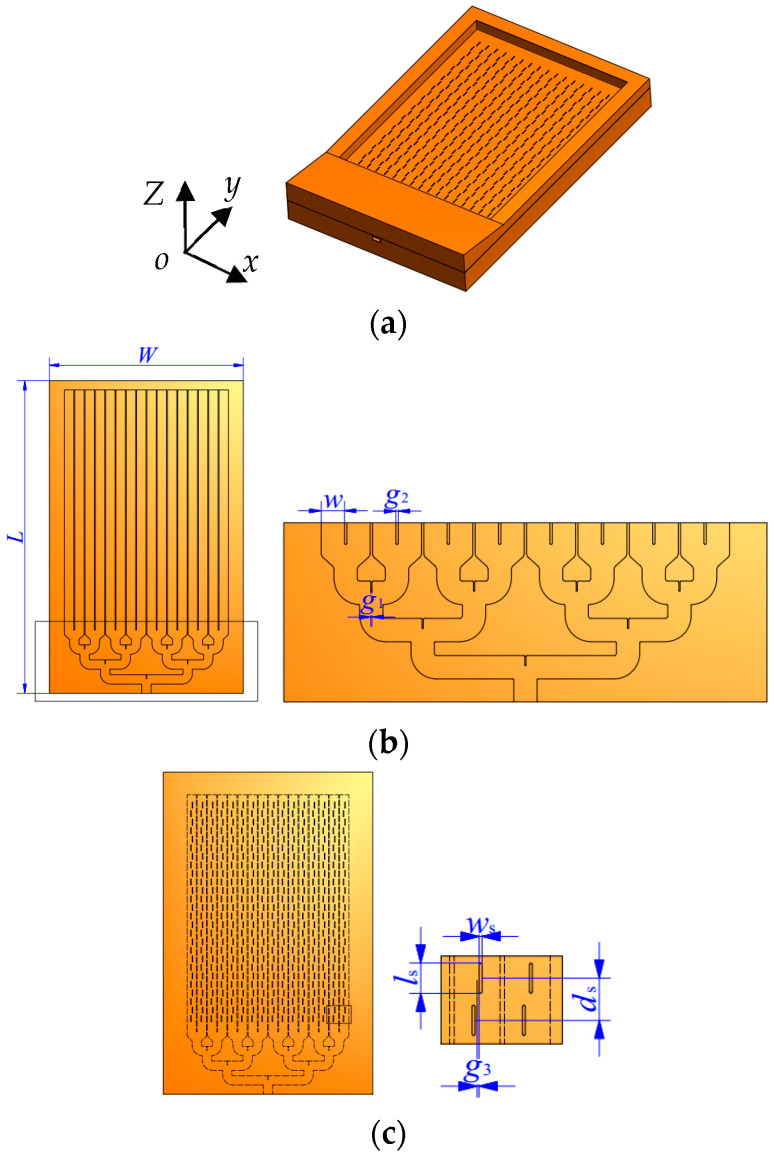
Configuration of the designed high-gain slot array antenna: (**a**) 3D view; (**b**) the substratum layer; (**c**) the superstratum layer.

**Figure 2 micromachines-14-01947-f002:**
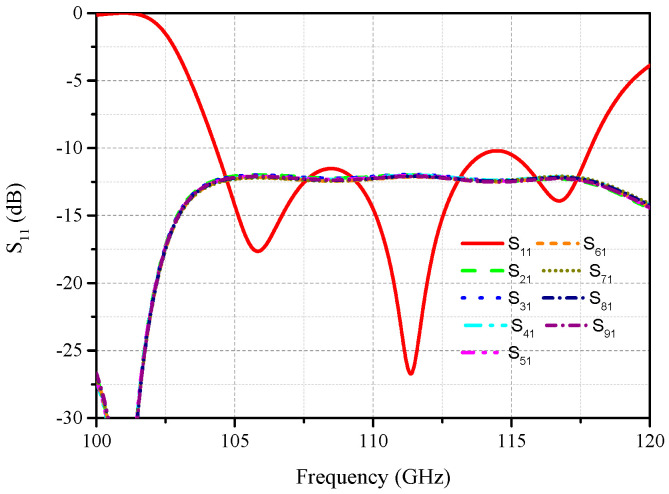
Simulated S_11_ results of the power divider (*w* = 2.03 mm, *g*_1_ = 0.1 mm, *g*_2_ = 0.2 mm).

**Figure 3 micromachines-14-01947-f003:**
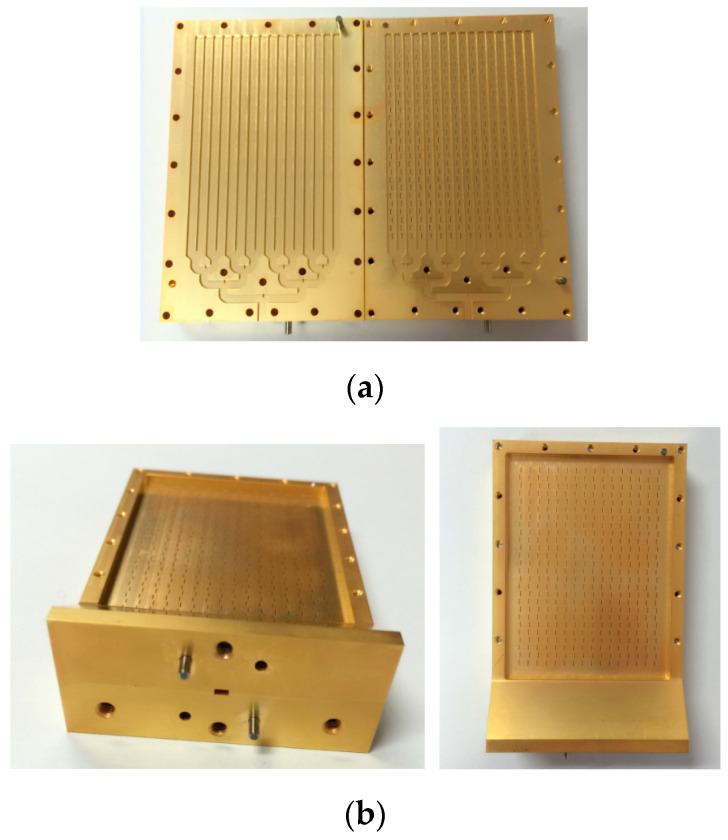
Photo of the fabricated slot array antenna: (**a**) top view; (**b**) side view.

**Figure 4 micromachines-14-01947-f004:**
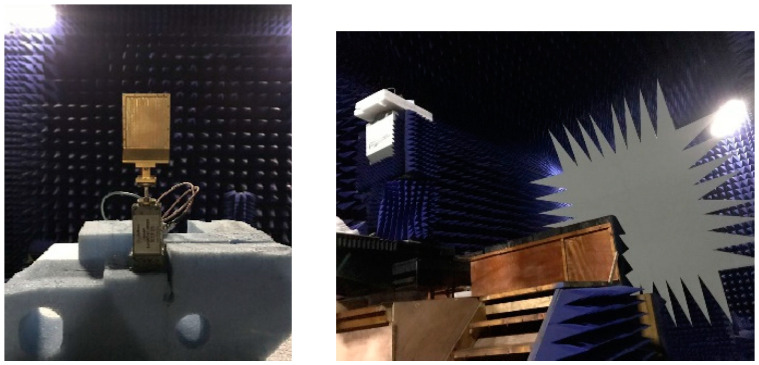
Photo of the test environment of the fabricated antenna.

**Figure 5 micromachines-14-01947-f005:**
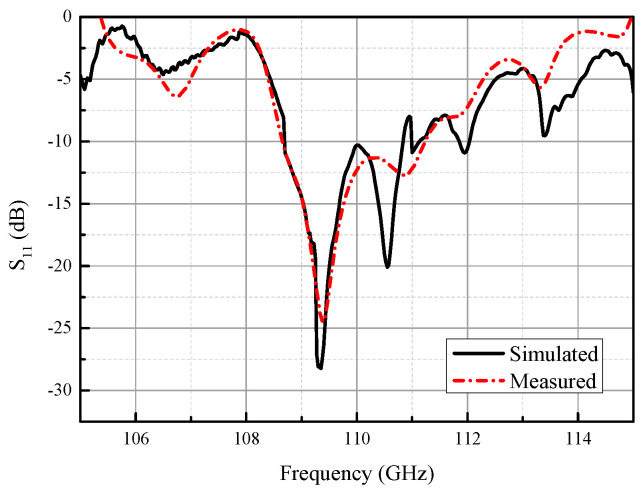
Simulated and measured S_11_ of the proposed slot array antenna (*w*_s_ = 0.13 mm, *l*_s_ = 1.35 mm, *d*_s_ = 1.87 mm, *g*_3_ = 0.15 mm).

**Figure 6 micromachines-14-01947-f006:**
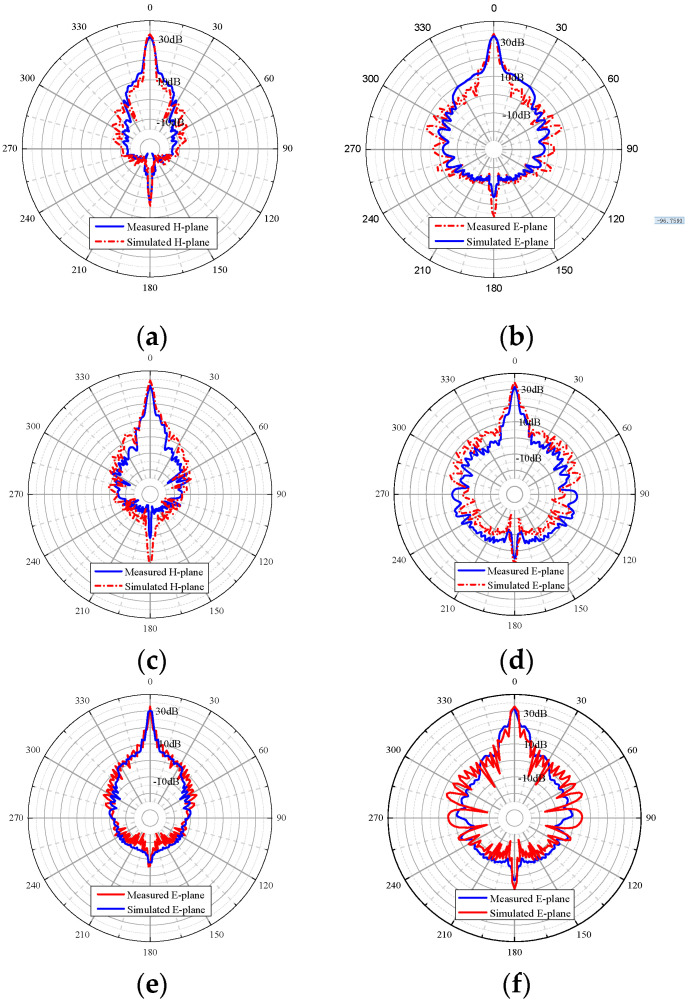
Measured and simulated normalized radiation patterns of the proposed antenna at different frequencies: (**a**) 109 GHz, *H*-plane; (**b**) 109 GHz, *E*-plane; (**c**) 110 GHz, *H*-plane; (**d**) 110 GHz, *E*-plane; (**e**) 111 GHz, *H*-plane; (**f**) 111 GHz, *E*-plane.

**Table 1 micromachines-14-01947-t001:** Comparison of performance among antennas working at millimeter wave.

*f*_0_ (GHz)	Size	Peak Gain (dBi)	Tech.	Ref.
136–144	*π*/2 (4.2*λ*_0_)^2^	23.4	lens	[5]
75–110	2.9*λ*_0_ *×* 1.4*λ*_0_	10	silicon	[6]
94.2–101.8	10*λ*_0_ *×* 6.7*λ*_0_ *×* 0.68*λ*_0_	19	PCB	[7]
87–101	10*λ*_0_ *×* 6.7*λ*_0_	23.8	LTCC	[8]
90–96.5	3.13*λ*_0_ *×* 3.13*λ*_0_	13.3	LTCC	[9]
80–102	6.5*λ*_0_ *×* 6.5*λ*_0_ *×* 0.34*λ*_0_	25.3	PCB	[10]
71–86	27*λ*_0_ *×* 27*λ*_0_ *×* 1.6*λ*_0_	37	chemical plate dafter CNC	[11]
84–104	7.1*λ*_0_ *×* 7.1*λ*_0_	27.7	micro-maching	[13]
95.1–98	11.5*λ*_0_ *×* 11*λ*_0_ *×* 2.8*λ*_0_	18.1	CNC	[14]
110	25*λ*_0_ *×* 14*λ*_0_	32	CNC	This work

## Data Availability

The research data is available when contact with the authors.

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
