# Peer review of "High Gain Slot Array Antenna at 110 GHz Based on Computer Numerical Control"

_micromachines, 2023, doi:10.3390/mi14101947_

Round 1

Reviewer 1 Report

This manuscript presents a 110-GHz slot antenna array by using CNC technology. Full-wave simulated and measured results are consistent with each other.  Here I have some minor comments.

1) It is seen from Fig. 2 that there has an additional insertion loss of around 3 dB, compared to an ideal 8-way power divider. It is suggested to give an explaination. 

2) In Table 1, it is mentioned that the gain of the antennas reported in [7] and [8] are much lower, compared to the proposed one in this manuscript. I think the main reason is that the array scales of those arrays are much small. As for the ones in [7] and [8], can the array aperture be extended? If YES, then I think it is not fair to simply compare the gain with different apreture. 

Author Response

Reply to Comments of Reviewer: 1

This manuscript presents a 110-GHz slot antenna array by using CNC technology. Full-wave simulated and measured results are consistent with each other.  Here I have some minor comments.

<Comment 1>: It is seen from Fig. 2 that there has an additional insertion loss of around 3 dB, compared to an ideal 8-way power divider. It is suggested to give an explanation. 

Response: Thanks for your suggestion. The feeding network is combined by a third-order power divider and a compact power divider because of the tightly arranged waveguide structure. Actually, the power divider is fourth order, so the insertion loss is around 12 dB. The explanation is also added in the revised manuscript.

<Comment 2>:  In Table 1, it is mentioned that the gain of the antennas reported in [7] and [8] are much lower, compared to the proposed one in this manuscript. I think the main reason is that the array scales of those arrays are much small. As for the ones in [7] and [8], can the array aperture be extended? If YES, then I think it is not fair to simply compare the gain with different aperture. 

Response: Thanks for this comment. The antennas designed in [7] and [8] have smaller aperture size than ours. If the array elements are added and the effective aperture size is extended, the gain value will be enhanced to a certain extend because of the much more complex structures with multi layers which would face inevitable energy loss.

Reviewer 2 Report

Dear authors, thank you for interesting studies of high gain and low loss antenna arrays operating at 110 GHz. The manuscript is well written and clear indicates that the network for 8 elements is suitable for high gin technology. The only that I would recommend adding some discussion on the comparison of presented technology with others such as micromaching of Si etc. That will be advantages and disadvantages

Author Response

Reply to Comments of Reviewer: 2

<Comment 1>: Dear authors, thank you for interesting studies of high gain and low loss antenna arrays operating at 110 GHz. The manuscript is well written and clear indicates that the network for 8 elements is suitable for high gin technology. The only that I would recommend adding some discussion on the comparison of presented technology with others such as micromaching of Si etc. That will be advantages and disadvantages.

Response: Thanks for your useful suggestion. The discuss about the Ref. [14] is added in the revised manuscript.
